

# Multi-task snake optimization algorithm for global optimization and planar kinematic arm control problem

Qingrui Li[1], Yongquan Zhou[1,2] and Qifang Luo[1,2]

[1] College of Artificial Intelligence, Guangxi Minzu University, Nanning, Guangxi, China
[2] Guangxi Key Laboratories of Hybrid Computation and IC Design Analysis, Nanning, Guangxi, China

## ABSTRACT

Multi-task optimization (MTO) algorithms aim to simultaneously solve multiple optimization tasks. Addressing issues such as limited optimization precision and high computational costs in existing MTO algorithms, this article proposes a multi-task snake optimization (MTSO) algorithm. The MTSO algorithm operates in two phases: first, independently handling each optimization problem; second, transferring knowledge. Knowledge transfer is determined by the probability of knowledge transfer and the selection probability of elite individuals. Based on this decision, the algorithm either transfers elite knowledge from other tasks or updates the current task through self-perturbation. Experimental results indicate that, compared to other advanced MTO algorithms, the proposed algorithm achieves the most accurate solutions on multitask benchmark functions, the five-task and 10-task planar kinematic arm control problems, the multitask robot gripper problem, and the multitask car side-impact design problem. The code and data for this article can be obtained from: https://doi.org/10.5281/zenodo.14197420.

## INTRODUCTION

Nowadays, multi-task optimization (MTO) has emerged as a new research direction in the field of optimization. MTO aims to simultaneously solve multiple optimization problem (*Osaba et al., 2022*), with each problem corresponding to a task. In real-life scenarios, there exists a variety of optimization problems, some of which are almost unrelated, while others have more or less intrinsic connections. Can the information between interrelated tasks be leveraged to achieve better results than individual optimization? In response to such questions, MTO algorithms have been developed. MTO focuses on how to simultaneously address multiple optimization problems, assuming similarities between problems, such as sharing the same optimal domain, landscape trends, *etc*. It harnesses the inherent parallelism of a population to solve a series of tasks simultaneously, thereby creating pathways for skill transfer between them (*Gupta et al., 2022*). Since its inception, MTO has been applied in various domains (*Gupta et al., 2022*), including multitask high-dimensional function optimization (*Gupta, Ong & Feng, 2015*), multitask large-scale

Corresponding author
Yongquan Zhou,
zhouyongquan@gxmzu.edu.cn

multi-objective optimization (*Liu et al., 2022*), multitask constraint optimization (*Xing, Gong & Li, 2023*), multitask engineering optimization (*Cheng et al., 2017*), multitask vehicle routing optimization (*Cai, Wu & Fang, 2024*), multitask multi-objective pollution-routing problem (*Rauniyar, Nath & Muhuri, 2019*), multitask power system scheduling (*Qiao et al., 2022*), and others.

*Gupta, Ong & Feng (2015)* initially proposed a multifactorial evolutionary algorithm (MFEA) for addressing multitask problems. *Xing, Gong & Li (2023)* introduced an archive-based adaptive MFEA. *Cai, Wu & Fang (2024)* proposed a dual-assisted evolutionary MTO algorithm. *Xu, Qin & Xia (2021)* introduced an adaptive evolutionary MTO algorithm (AEMTO) with adaptive knowledge transfer. *Jiang et al. (2023)* presented a block-level knowledge transfer multitask evolutionary algorithm. *Li, Gong & Li (2023a)* proposed an evolutionary MTO algorithm utilizing knowledge-guided external sampling strategies. *Yang et al. (2023)* employed surrogate models to assist knowledge transfer (KT) between tasks, effectively alleviating negative KT. *Jiang et al. (2022)* proposed a dual-objective KT framework in evolutionary multitask optimization algorithms, effectively utilizing knowledge between tasks. *Ji et al. (2021a)* introduced a dual-surrogate-assisted cooperative particle swarm optimization (PSO) algorithm. *Ji et al. (2021b)* proposed a multi-surrogate-assisted multitask PSO algorithm. *Wu & Tan (2020)* introduced a multitask genetic algorithm (GA) for fuzzy system optimization. *Tuo et al. (2023)* proposed a multitask harmony search (HS) algorithm. *Li, Gong & Li (2023b)* introduced an adaptive solver multitask evolutionary framework. *Ma et al. (2023)* enhanced evolutionary multitask optimization by utilizing KT between tasks and improved evolutionary operators. *Li et al. (2024)* proposed a multitask, multi-objective differential evolution (DE) gene selection algorithm for tumor identification. *Yuan et al. (2016)* utilized permutation-based unified representation and level-based selection to enhance the original MFEA. *Bali et al. (2017)* proposed a linearized domain-adaptive MFEA. *Liaw & Ting (2017)* introduced a symbiotic evolutionary bio-community algorithm. *Ding et al. (2017)* proposed a generalized MFEA. *Feng et al. (2018)* used autoencoders in evolutionary multitasking to explicitly transfer knowledge across tasks. *Wen & Ting (2017)* proposed a MFEA with resource reallocation. *Dao, Tam & Binh (2024)* proposed an evolutionary multitask algorithm to maximize the network lifetime of wireless sensor networks. *Li, Gong & Li (2022)* proposed an evolutionary constrained multitasking optimization algorithm.

Currently, there are mainly two types of multitask algorithm frameworks. One type is based on multi-factor approaches. The other type is based on multi-population approaches (*Wei et al., 2021*). Most multi-task optimization approaches are based on evolutionary algorithms, though some utilize swarm intelligence (SI) methods such as PSO.

The snake optimization (SO) algorithm is a recently proposed population-based bio-inspired algorithm. It has been applied to a variety of optimization problems, such as disease diagnosis (*Khurma et al., 2023*), real-world engineering problems (*Yao et al., 2023*), feature selection (*Al-Shourbaji et al., 2022*), DDoS attack detection (*Aljebreen et al., 2023*), complex signal analysis (*Li et al., 2023a*), workflow scheduling in cloud computing (*Li et al., 2023b*), fingerprint localization (*Zheng et al., 2023*), filter optimization design (*Janjanam, Saha & Kar, 2022*), load change response in photovoltaic (*Mohammed &*

*Mekhilef, 2023*), neural network optimization (*Yan & Razmjooy, 2023*), photovoltaic parameter extraction (*Belabbes et al., 2023*), energy optimization in hybrid energy storage systems (*Wang et al., 2023*) and other complex optimization problems. The SO algorithm has achieved promising results in these optimization problems, demonstrating its effectiveness in handling complex optimization challenges.

The main contributions of this article are as follows:

1. A new multi-task SO (MTSO) algorithm is proposed.

2. The effectiveness and accuracy of the MTSO through numerical experiments and comparisons with other advanced MTO algorithms.

3. The MTSO algorithm was applied to planar kinematic arm control problems (PKACP), robot gripper design problem, and car side-impact design problems. Its superior performance was demonstrated through comparisons with other advanced MTO algorithms.

The structure of this article is as follows: "Preliminary" primarily introduces background knowledge of multitask optimization problem and the fundamental SO Algorithm. "The Proposed Algorithm" outlines the proposed algorithm framework. "Experimental Results and Analysis" is to discuss and analyze the experimental results. "Testing with Real-World Engineering Problems" focuses on the application of the MTSO algorithm to real-world engineering problems. Finally, "Conclusion" provides a summary and outlines future work.

# PRELIMINARY

## Multitask optimization problem

Assuming the need to concurrently optimize $K$ tasks, this article posits that all $K$ tasks are minimal optimization problems. Let $T_i$ ($i = 1, 2, \dots, K$) denote the $i$-th task. An MTO problem can be defined as follows:

$$\begin{cases} \{x_1, x_2, \dots, x_K\} = \arg\min\{f_1(x), f_2(x), \dots, f_K(x)\} \\ x_i \in \Omega_i, i = 1, 2, \dots, K \end{cases} \tag{1}$$

where, $f_i$ represents the fitness function of task $T_i$, $x_i$ denotes the solution of task $T_i$, and $\Omega_i$ denotes the search space for task $T_i$.

## Snake optimization algorithm

The SO algorithm is a recently proposed population-based bio-inspired algorithm. The quantity of food serves as the boundary between exploration and exploitation. In the exploration stage, when the food is scarce, snakes search the entire search space for sustenance. As the food quantity reaches a threshold, the algorithm transitions into the exploitation stage, further divided into three modes: consuming existing prey, combat, and mating. The key factor determining the mode is the temperature. For detailed information about the algorithm, please refer to *Hashim & Hussien (2022)*.

### Abbreviations of professional nomenclature and their meanings

This section mainly summarizes the abbreviations of professional nomenclature mentioned in the article and their meanings, as shown in Table S1.

## THE PROPOSED ALGORITHM

### Motivation

There is a relative scarcity of multitask algorithms based on SI. The SO algorithm is a recently proposed bio-inspired optimization algorithm based on SI. It exhibits excellent performance in solving high-dimensional, nonlinear complex optimization problems. Therefore, introducing a multitask version of the SO algorithm to address multitask optimization problems holds considerable research value.

### The algorithm framework

The multitask algorithm proposed in this article is based on multi-population approaches and consists of two phases: the independent optimization phase using the SO algorithm, and the knowledge transfer (KT) phase.

**Independent optimization phase:** For an optimization problem with n tasks, a separate sub-population is assigned to each task. Each task is independently optimized, and the top 1/5 elite individuals based on fitness are selected and stored in the elite repository. Random numbers r1 and r2 are generated.

**Knowledge transfer phase:** KT is primarily controlled by the knowledge transfer probability (RMP) and elite individual selection probability (R1). RMP is set as a constant value of 0.5, and R1 is set as a constant value of 0.95. Knowledge transfer is divided into two parts: KT between different tasks and self-knowledge transfer within tasks, namely, random perturbation of the solution. (**Note:** Normalization of individuals is performed before knowledge transfer).

The normalization process is outlined in Eq. (2). The pseudo code of the algorithm is provided in Table S2.

$$X_{ij}^* = \frac{X_{ij} - Lb_j}{Ub_j - Lb_j} \tag{2}$$

where $X_{ij}^*$ and $X_{ij}$ are, respectively, the normalized and original j-th dimension of the *i*-th individual, and $Lb_j$ and $Ub_j$ are, respectively, the lower and upper bounds of the *j*-th dimension.

If random number r1 is less than RMP and random number r2 is less than R1, inter-task knowledge transfer occurs. A random task is selected as the source task, and knowledge is transferred from elite individuals of the source task to the target task. If r1 is less than RMP and r2 is greater than R1, random perturbation is applied to the worst-performing individual of the target task. If r1 is greater than RMP, the reverse learning through lens imaging strategy is used to evaluate and select the best-performing individuals among the reversed individuals of the target task. These selected individuals are retained for the next iteration. The principle of reverse learning through lens imaging is as follows:

$$x_j^{'*} = \frac{L_j + U_j}{2} + \frac{L_j + U_j}{2k} - \frac{x_j^*}{k} \tag{3}$$

where $k = \left(1 + \left(\frac{t}{T}\right)^{0.5}\right)^{10} x_j^{'*}$ and $x_j^*$ are the $j$-th dimensional components of $x^{'*}$ and $x^*$, and $U_j$ and $L_j$ are the $j$-th dimensional upper and lower bounds of the decision variables.

## Algorithm complexity analysis

### Computational complexity

Assume K is the number of tasks, T is the number of iterations, N is the population size for each task, and d is the maximum dimension of decision variables. The solution phase of the algorithm consists of two stages: the independent solving stage and the knowledge transfer stage. Independent solving stage: The overall time complexity for the independent solving stage is O(K * N * d). Knowledge transfer stage: The time complexity for this stage is O(K * N/5 * d). Combining both stages, the total time complexity of the algorithm is: T * [O(K * N * d) + O(K * N/5 * d)] = O(T * K * N * d).

### Space complexity

The space complexity of the algorithm primarily depends on four factors: the number of tasks K, the population size N, the dimensionality of each individual d, and auxiliary data structures. The space requirements for these components are as follows: 1. Storing the individuals: O(K * N * d). 2. Storing the fitness values: O(K * N). 3. Storing elite knowledge: O(K * N/5 * d). 4. Storing normalized individuals: O(K * N * d). Adding these together, the total space complexity is: O(K * N * d).

## Detection and prevention of negative knowledge transfer

Negative knowledge transfer occurs when the optimization results after knowledge transfer are worse than those achieved without it. To detect and prevent negative knowledge transfer, a comparison mechanism is implemented during the knowledge transfer stage. This mechanism compares the fitness values before and after knowledge transfer. If the results obtained after transferring knowledge from other tasks degrade, the transfer is discarded. Instead, the solution from the independent optimization stage is chosen as the optimal solution for that iteration.

## EXPERIMENTAL RESULTS AND ANALYSIS

The simulation environment for this experiment is MATLAB 2018a (The MathWorks, Natick, NY, USA) running on an Intel Core i3-6100 computer with a CPU frequency of 3.700 GHz and 8 GB of RAM. The algorithm parameters are set as shown in Table 1. In the experiments, each task corresponds to a population, and each population is randomly generated within the predefined upper and lower bounds of the respective task. To ensure that the performance of the algorithm is not affected by the initial population, the same initial population is used for each task across all comparison algorithms.

**Table 1 Parameter settings.**

| Symbol | Meaning | Parameters value |
|---|---|---|
| popSize | Population size | popSize = 100 |
| nGen = 500 | Number of generations | nGen = 500 |
| RMP | Knowledge transfer probability | RMP = 0.5 |
| nRepeat | Number of repeats | nRepeat = 20 |
| dim | Dimension of the problem | dim = 30 |
| R1 | Elite individual selection probability | R1 = 0.95 |

## Benchmark function testing

In this section, the effectiveness of the MTSO algorithm is tested using nine sets of multitask problems constructed from seven multi-modal functions, sourced from literature (*Yang et al., 2023*). To highlight the performance of the MTSO algorithm, the following algorithms are selected for comparison: multi-factor evolutionary algorithm (MFEA) (*Gupta, Ong & Feng, 2015*), surrogate assisted evolutionary multitasking optimization algorithm (SAMTO) (*Yang et al., 2023*), multi-task evolutionary algorithm (MTEA) (*Wu & Tan, 2020*), multi-factor evolutionary algorithm with level-based selection (MFEALBS) (*Yuan et al., 2016*), linearized domain adaptive multi-factor evolutionary algorithm (LDAMFEA) (*Bali et al., 2017*), evolutionary bio-community symbiotic multi-task algorithm (EBSGA) (*Liaw & Ting, 2017*), generalized multi-factor evolutionary algorithm (GMFEA) (*Ding et al., 2017*), evolutionary multitasking *via* explicit autoencoding (EMTEA) (*Feng et al., 2018*), and multi-factor evolutionary algorithm with resource reallocation (MFEARR) (*Wen & Ting, 2017*). Each algorithm is independently run 20 times for comparison. The optimal values for the seven multi-modal functions selected in this article are all 0, with a dimensionality of 30. Basic information about the test functions is shown in Table S3, and the nine sets of multitask optimization problems and their task similarities are presented in Table 2.

## Results of benchmark function testing

In this section, all experiments were independently run 20 times, and the mean and standard deviation (Std) were used as evaluation criteria. The best-performing results are indicated in bold. The experimental results for the benchmark functions are shown in Table S4, where "run time" represents the time taken to solve a set of tasks simultaneously, and the "$p$-value" is the result of the Wilcoxon rank-sum test. A $p$-value less than 0.05 indicates a significant difference between the algorithms. The convergence curve plots for the nine sets of experiments are shown in Fig. S1, the error bar chart is shown in Fig. S2.

## Discussion and analysis of benchmark function testing

Based on Table S4 and Fig. S1, the MTSO algorithm achieved the optimal value of 0 in seven out of nine experiments. Only the third and ninth experiments showed suboptimal performance. Due to the no free lunch theorem, no algorithm can solve all optimization problems. MTSO performs poorly on Task 2 of Test Cases 3 and 9, particularly in solving

**Table 2 Multitask problem construction.**

| Test combinations | Task 1 | Task 2 | Inter-task similarity |
|---|---|---|---|
| 1 | 1 (Griewank) | 2 (Rastrigin) | 1.000 |
| 2 | 3 (Ackley) | 2 (Rastrigin) | 0.2261 |
| 3 | 3 (Ackley) | 4 (Schwefel) | 0.0002 |
| 4 | 2 (Rastrigin) | 5 (Sphere) | 0.2154 |
| 5 | 3 (Ackley) | 7 (Rosenbrock) | 0.8670 |
| 6 | 3 (Ackley) | 6 (Weierstrass) | 0.0725 |
| 7 | 2 (Rastrigin) | 7 (Rosenbrock) | 0.9434 |
| 8 | 1 (Griewank) | 6 (Weierstrass) | 0.3669 |
| 9 | 2 (Rastrigin) | 4 (Schwefel) | 0.0016 |

the Schwefel function, where it fails to find the global optimum. This is due to the function's characteristic of having numerous local optima, making it difficult to find the global optimum. Additionally, due to the low similarity between tasks (*Wu & Tan, 2020*), the knowledge from Task 1 is not effective in assisting Task 2. As a result, the algorithm becomes trapped in a local optimum, leading to poor performance of MTSO in solving Test Cases 3 and 9. The experimental results demonstrate that compared to other advanced MTO algorithms, the MTSO algorithm exhibits stronger competitiveness in handling multitask problems.

## Noisy perturbed benchmark function testing

To better evaluate the stability and effectiveness of the proposed algorithm, this section constructs nine sets of perturbed benchmark functions based on the functions from "Benchmark Function Testing". The construction method involves adding 10 to each function, *i.e.*, f = f + 10. The task construction follows the same setup as in Table 2. The optimal value of each task has become 10.

## Perturbed benchmark function testing results

All experiments in this section were run independently 20 times. The experimental results for the benchmark functions are shown in Table S5. The convergence curves for the nine sets of experiments are presented in Fig. S3, and the error bar chart is shown in Fig. S4.

## Population size analysis

This section analyzes the population size of the proposed algorithm. The selected population sizes are 30, 50, and 100. The experiments are conducted using the multitasking test functions constructed earlier, with each population size independently run 20 times. The algorithm's parameters, except for the population size, remain the same across all experiments. The results are then analyzed using the Friedman test to obtain the average ranking. A smaller average ranking indicates a better algorithm performance. The Friedman ranking of different population sizes are shown in Table S6.

Based on the results in Table S6, we observe that the average rankings are identical for certain test cases, indicating that population size has minimal impact on the relative performance of the algorithm. However, considering all test cases comprehensively, the algorithm achieves the best results when the population size is set to 100. Therefore, we select a population size of 100.

## Parameter sensitivity analysis

This section analyzes the parameters of the proposed algorithm. Twelve different parameter combinations were tested using the multitasking test functions constructed earlier. Each combination was independently run 20 times. The results were then analyzed using the Friedman test to obtain the average ranking, where a smaller average ranking indicates better algorithm performance. Friedman ranking with different parameter settings in Table S7.

Based on the results in Table S7, we observe that the average rankings are identical for certain test cases, indicating that the values of the RMP and R1 parameters have minimal impact on the relative performance of the algorithm. However, considering all test cases comprehensively, the algorithm achieves the best results when RMP = 0.5 and R1 = 0.95, followed by RMP = 0.3 and R1 = 0.85.

## Analysis of knowledge transfer scale

This section analyzes the scale of knowledge transfer in the proposed algorithm, specifically examining how much knowledge is transferred from the source task to the target task. The selected knowledge transfer scales are set as 1/10, 1/5, and 1/2 of the population size. Experiments were conducted on the multi-task test functions constructed earlier, with each population running independently 20 times. The obtained results were evaluated using Friedman tests to calculate average rankings, where a smaller average ranking indicates better algorithm performance. Friedman ranking of different knowledge transfer scale in Table S8.

Based on the results in Table S8, we observe that the average rankings are identical for many test cases, indicating that the scale of knowledge transfer has minimal impact on algorithm performance. This is because only the optimal knowledge is transferred, and transferring the knowledge of the best individual is sufficient to guide optimization in the target task. However, when considering all test cases comprehensively, we find that the total average ranking is smallest when the knowledge transfer scale is 1/5 of the population size, resulting in the best algorithm performance. Therefore, we select 1/5 of the population as elite individuals for knowledge transfer.

## Analysis of knowledge utilization rate

This section introduces quantitative metrics to measure the quality and effectiveness of knowledge transfer. A counter is implemented in the algorithm to track the fitness values before and after knowledge transfer in each iteration. If the fitness value of the individual improves (decreases for minimization problems) after knowledge transfer, the transfer is considered effective, and the counter is incremented by 1. Instances without knowledge

transfer are marked as −1. Let the total number of iterations be N, the effective transfer count be a, and the count of no transfer occurrences be b. The knowledge utilization rate is then calculated using the formula shown in Eq. (4).

The knowledge utilization rates for various test functions are presented in Table S9. We define each iteration of knowledge transfer as effective (1), ineffective (0), or not performed (−1), and plot the knowledge transfer results for each iteration as shown in Fig. S5.

From Fig. S5, it can be observed that knowledge transfer is effective during the early stages of iteration, aiding rapid convergence of the optimization task. However, in the later stages, knowledge transfer provides limited assistance in solving the optimization task.

$$R = a/(N - b) \tag{4}$$

## Convergence analysis of the algorithm

The proposed algorithm belongs to the category of swarm intelligence optimization algorithms, which are rooted in experimental science. Proving the convergence of swarm intelligence algorithms is often a complex and challenging task, making it difficult to analyze using traditional mathematical methods. Therefore, this study investigates the convergence and performance of the algorithm through numerical simulations and experimental validation. First, we define the objective functions, which are the benchmark test functions constructed earlier, and set the initial and termination conditions of the algorithm. In the numerical experiments, a counter is employed: if the result of the next iteration is the same as that of the previous iteration, the counter increments by one. When the counter reaches 100, the algorithm is considered to have reached a stable state and is deemed to have converged. The algorithm was tested under different parameter settings, and noise was added to the test functions. Through extensive experiments, we demonstrated that the proposed algorithm consistently reaches a stable state within a finite number of iterations. This is particularly evident in the algorithm's convergence curves and error bar plots, where the stable state of the algorithm is clearly observed.

## TESTING WITH REAL-WORLD ENGINEERING PROBLEMS

### Planar kinematic arm control problem

This section primarily focuses on applying the MTSO algorithm to solve the planar kinematic arm control problem (PKACP). The PKACP can be described as a practical industrial application. In our context, the robotic arms are viewed as components in an assembly line, performing precise and efficient tasks such as part assembly. Specifically, in body welding and component installation processes, multiple robotic arms need to accurately position parts on a vehicle body at designated locations for precise welding or installation. Each robotic arm represents a task, and the optimization of multiple robotic arms simultaneously delivering parts to specific locations constitutes a multi-task optimization problem. Figure S6 illustrates the schematic diagram of the PKACP for two tasks. This image is from *Jiang et al. (2023)*. To better evaluate the performance of the MTSO algorithm in solving real-world engineering problems, this article utilizes the algorithm to solve a PKACP with 5 and 10 tasks.

**Problem definition:** The multitask PKACP requires finding the optimal angles for all joints for each task (*i.e.*, $\alpha_1, \alpha_2, \ldots, \alpha_d$) to minimize the Euclidean distance between the arm tip (*i.e.*, $P_D$) and the target (*i.e.*, T) (*Jiang et al., 2023*). The objective function for the $i$ task is defined as follows:

$$f_i\left(\alpha_1, \alpha_2, \ldots \alpha_d, \left[L_i, \alpha_{i,\max}\right]\right) = \|P_D - T\| \tag{5}$$

where $L_i$ and $\alpha_{i,max}$ represent, respectively, the total length of the arm and the maximum range of angles for the $i$-th PKACP. Different tasks are created by taking different values of $L_i$ and $\alpha_{i,max}$. The values of $L_i$ and $\alpha_{i,max}$ are determined as follows: first, two-dimensional random samples are generated, then, according to the number of tasks ntasks, K-means clustering is used to cluster the samples into ntasks classes, where the first dimension of the cluster centers represents the value of $L_i$, and the second dimension represents the value of $\alpha_{i,max}$. The target position is set to [0.5, 0.5]. To better test the performance of the multi-task snake optimization algorithm in solving real-world problems, this article establishes multi-task PKACP with 5 and 10 tasks and solves them using the multi-task snake optimization algorithm. The number of joints (*i.e.*, dimensions) for each task is set to 5, 10, and 20, resulting in a total of six experimental groups.

## Experimental results of the PKACP

In this article, multi-task PKACP with five and 10 tasks were constructed, where the values of $L_i$ and $\alpha_{i,max}$ are randomly generated. For comparison purposes, the optimal values obtained for all tasks were averaged and compared with other advanced MTO algorithms. The selected comparison algorithms include: MFEA (*Gupta, Ong & Feng, 2015*), MTEA (*Wu & Tan, 2020*), EBSGA (*Liaw & Ting, 2017*), GMFEA (*Ding et al., 2017*), EMTEA (*Feng et al., 2018*), and MFEARR (*Wen & Ting, 2017*). Note that all the parameters for the comparison algorithms are sourced from their respective literature, and no modifications have been made in this article. All algorithms are independently run 20 times, and mean and Std are used as evaluation criteria. The experimental results are shown in Table S10. The convergence curves and error bars are illustrated in Figs. S7 to S12.

## Discussion and analysis of the PKACP

Based on Table S11 and Figs. S7 to S12, it can be concluded that the MTSO algorithm proposed in this article is more competitive in terms of both accuracy and convergence speed compared to other advanced multitask algorithms. As the number of tasks increases, the error decreases, demonstrating that the multi-task algorithm is suitable for handling a large number of tasks. However, as the dimensionality increases, the error also increases, indicating that the multi-task algorithm still has some shortcomings in handling high-dimensional problems. Although the MTSO algorithm achieves the smallest error in the PKACP, it does not have the best standard deviation, ranking third among all algorithms. The best standard deviation is achieved by the EMTEA. Overall, the MTSO algorithm performs the best when considering both accuracy and convergence speed.

### Robot gripper design problem

The objective of this optimization problem is to minimize the difference between the maximum and minimum forces exerted by the gripper, subject to the range of gripper end displacement (*Yin, Luo & Zhou, 2022*). There are seven continuous design variables (a, b, c, d, e, f, δ). Each robotic gripper represents a task, and this article addresses the simultaneous solution of two robotic gripper problems. The robotic gripper problem is subjected to seven distinct constraint conditions. Please refer to *Yin, Luo & Zhou (2022)* for detailed mathematical formulas.

The statistical results of MTSO and other comparison algorithms on the two-task robotic gripper problem are shown in Table S11. The convergence curves and error bars are illustrated in Fig. S13. According to Table S11 and Fig. S13, it can be concluded that our proposed algorithm achieved the best results.

### Car side impact design problem

The objective of the side-impact collision problem for automobiles is to minimize the total weight of the car using 11 mixed variables while maintaining standard safety performance (*Zhang et al., 2023*). Variables 8 and 9 are discrete, while the rest are continuous. It can be viewed as a mixed discrete and continuous mechanical optimization problem. Since the 8th and 9th variables are discrete, different tasks can be constructed by assigning different values to these variables. For Task 1, the 8th and 9th variables are set to 0.192, while for Task 2, the 8th and 9th variables are set to 0.345. Please refer to *Zhang et al. (2023)* for detailed mathematical formulas.

The statistical results of MTSO and other comparison algorithms on the two-task car side impact design problem are shown in Table S12. The convergence curves and error bars are illustrated in Fig. S14. According to Table S12 and Fig. S14, it can be concluded that our proposed algorithm achieved the best results.

## CONCLUSION

This article proposes a multitask version of the SO algorithm, based on the original SO algorithm, to simultaneously address multiple optimization problems. The MTSO algorithm determines whether to transfer elite knowledge from other tasks or to update the task's own perturbation through preset knowledge transfer probabilities RMP and elite individual selection probability R1. Through numerical experiments, the planar kinematic arm control problem, the robot gripper design problem, and the car side impact design problem have demonstrated the effectiveness of the MTSO algorithm and its ability to solve real-world engineering problems.

The MTSO algorithm proposed in this article is single-objective. Future work could focus on developing a multi-objective version. Additionally, while this article addresses negative knowledge transfer by discarding harmful knowledge, it does not fundamentally resolve the issue of negative knowledge transfer. Future research will aim to tackle the root causes of negative knowledge transfer. Improving the performance of the algorithm on high-dimensional problems is also a focus for future research.

### Funding

This work is supported by the National Natural Science Foundation of China under Grant No. U21A20464, 62066005 and the Innovation Project of Guangxi Graduate Education under Grant No. YCSW2024313. The funders had no role in study design, data collection and analysis, decision to publish, or preparation of the manuscript.

### Grant Disclosures

The following grant information was disclosed by the authors:
National Natural Science Foundation of China: U21A20464, 62066005.
Innovation Project of Guangxi Graduate Education: YCSW2024313.

### Competing Interests

The authors declare that they have no competing interests.

### Author Contributions

- Qingrui Li conceived and designed the experiments, performed the experiments, analyzed the data, performed the computation work, prepared figures and/or tables, authored or reviewed drafts of the article, and approved the final draft.
- Yongquan Zhou conceived and designed the experiments, authored or reviewed drafts of the article, and approved the final draft.
- Qifang Luo analyzed the data, authored or reviewed drafts of the article, and approved the final draft.

### Data Availability

The code and data are available at GitHub and Zenodo:
- https://github.com/lqr123456/MTSO-PKACP.git
- lqr123456. (2024). lqr123456/MTSO-PKACP: v1.0.0 (v1.0.0). Zenodo. https://doi.org/10.5281/zenodo.14197420.

### Supplemental Information

Supplemental information for this article can be found online at http://dx.doi.org/10.7717/peerj-cs.2688#supplemental-information.

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
