# Peer review of "Multi-task snake optimization algorithm for global optimization and planar kinematic arm control problem"

_PeerJ Computer Science, doi:10.7717/peerj-cs.2688_

## Round 0.1 · original submission · Major Revisions

Both reviewers have provided detailed comments - please address them thoroughly in a revision.

·

Basic reporting

• The manuscript requires improvement in English language usage throughout. Specific instances requiring attention occur on lines 21, 78, and 121, where sentence structure and word choice could be enhanced for clarity. (Line 21: "This decision 21 determines whether to transfer elite knowledge from other tasks or to update tasks through self-22 perturbation." Problem: This sentence structure is awkward and could be clearer. Suggested revision: "Based on this decision, the algorithm either transfers elite knowledge from other tasks or updates the current task through self-perturbation."
Line 78: "For most multitask algorithms, the underlying 78 algorithms are typically evolutionary algorithms, and some utilize SI algorithms such as PSO." Problem: The sentence is redundant with "algorithms" appearing twice and has awkward phrasing. Suggested revision: "Most multi-task optimization approaches are based on evolutionary algorithms, though some utilize swarm intelligence (SI) methods such as PSO."
Line 121: "The next step is to find the best male and female individuals as well as the location of food." Problem: The sentence is too abrupt and lacks connection to the previous context. Suggested revision: "After population initialization, the algorithm identifies the best male and female individuals and determines the location of food.")
• Convergence plots should include error bars to better represent the statistical significance of the results across multiple runs.

Experimental design

1• The paper may include experimental results with noisy objective functions to demonstrate the algorithm's robustness.
• Testing should be extended to include constrained optimization problems, as these are common in real-world applications.
• Results with varying population sizes should be presented to guide users in selecting appropriate population parameters.
• Additional real-world applications beyond PKACP should be included to demonstrate broader applicability.
2. Theoretical Foundation and Analysis
• The paper would benefit from a dedicated section analyzing the theoretical convergence properties of MTSO, including formal proofs where possible.
• Please include a detailed computational complexity analysis comparing MTSO with existing algorithms to better understand its efficiency advantages.
• Mathematical justification for the algorithm's performance characteristics should be provided, particularly regarding the knowledge transfer mechanism.
• The manuscript needs an analysis of algorithm stability under different conditions, including proof of convergence under various scenarios.

3. Parameter Selection and Sensitivity
• A comprehensive sensitivity analysis for key parameters (RMP and R1) should be included to understand their impact on algorithm performance.
• The paper should present experimental results with different parameter settings to demonstrate robustness.
• Please provide clear guidelines for parameter selection based on different problem characteristics and complexity levels.
• The chosen values (RMP = 0.5, R1 = 0.95) need empirical justification through comparative experiments with alternative values.

4. Knowledge Transfer Mechanism
• The paper should develop and describe specific mechanisms to detect and prevent negative knowledge transfer between tasks.
• Additional experiments are needed to demonstrate the effectiveness of knowledge transfer in different scenarios, particularly when tasks have varying degrees of similarity.
• Please include a detailed analysis of conditions under which knowledge transfer helps versus hinders optimization performance.
• The manuscript should provide quantitative metrics for measuring knowledge transfer quality and effectiveness.

Validity of the findings

Q1: How was the elite repository size (top 1/5) determined? Please explain if any experimentation was conducted with different proportions and why this specific ratio was chosen.

Q2: The choice of reverse learning through lens imaging strategy needs justification. Why was this specific strategy selected over other potential approaches?
Q3: Please explain how the algorithm handles tasks with significantly different scales or characteristics. Are there any special normalization or adaptation mechanisms beyond what is currently described?

Q4: Could you provide comparative computational times across all algorithms tested? This information is crucial for practical implementation considerations.
Limited discussion of why MTSO performs worse in specific test cases (particularly tests 3 and 9)

Q5: The initialization process for populations in each task needs to be detailed. How were the initial populations generated and was this process consistent across all compared algorithms?

Q6: The relatively poor performance on test cases 3 and 9 requires explanation. What specific characteristics of these test cases caused difficulties for MTSO?

Q7: Could you provide an analysis of memory requirements for large-scale problems? This is crucial for practical applications.

Additional comments

I) High-Level Overview
This manuscript introduces a Multi-Task Snake Optimization (MTSO) algorithm, extending the capabilities of the existing Snake Optimization (SO) algorithm to handle multiple optimization tasks simultaneously. The core contribution lies in developing a two-phase optimization approach: independent optimization followed by knowledge transfer between tasks.
The work addresses a significant challenge in the field of optimization - the need for efficient algorithms that can handle multiple tasks concurrently while maintaining high precision and reasonable computational costs. The authors validate their approach through extensive experimentation, comparing MTSO against nine state-of-the-art multi-task optimization algorithms using both benchmark functions and a practical engineering application (the Planar Kinematic Arm Control Problem).
Strengths:
1. Methodological Innovation and Implementation
• Novel integration of Snake Optimization (SO) with multi-task optimization principles
• Clear and well-structured two-phase approach (independent optimization and knowledge transfer)
• Efficient implementation of knowledge transfer mechanisms with adaptive probability controls
• Thoughtful consideration of normalization in the knowledge transfer process to handle different task bounds

2. Experimental Design and Validation
• Comprehensive validation using 9 sets of benchmark functions
• Thorough comparison against 9 state-of-the-art algorithms in the field
• Robust statistical analysis with 20 independent runs for each experiment
• Clear presentation of both mean and standard deviation metrics for all comparisons
• Well-designed escalating complexity tests in the PKACP case study (5 and 10 tasks, with varying dimensions)

3. Results and Performance
• Superior accuracy demonstrated in benchmark functions (achieving optimal values in 7 out of 9 test cases)
• Consistently better performance in PKACP experiments across different dimensions and task numbers
• Competitive convergence speed compared to existing algorithms
• Robust performance across different problem scales and complexities

4. Documentation and Reproducibility
• Well-documented algorithm implementation
• Clear pseudocode presentation for both SO and MTSO algorithms
• Publicly available code and data through GitHub repository
• Detailed parameter settings and experimental conditions provided

5. Practical Application
• Strong demonstration of real-world applicability through the PKACP case study
• Systematic testing with increasing complexity (5 to 10 tasks)
• Practical consideration of dimensional scalability (5, 10, and 20 dimensions)
• Clear evidence of algorithm's effectiveness in handling practical engineering problems

6. Technical Writing and Presentation
• Logical organization of content
• Clear presentation of experimental results through well-designed tables and figures
• Effective use of convergence curves to illustrate algorithm performance
• Comprehensive explanation of the algorithm's components and workflow

7. Future Research Direction
• Clear identification of limitations and future work
• Thoughtful consideration of multi-objective extensions
• Recognition of the need to address negative knowledge transfer
• Acknowledgment of high-dimensional performance challenges

Reviewer 2 ·

Basic reporting

1) This paper is about multi-task optimization, where authors are proposing a new multitask optimization algorithm called MTSO. The algorithm handles optimization tasks separately then proceed with a knowledge transfer phase and then use a decision mechanism.
The paper starts by an introduction in which authors briefly review the main MTO contributions, then a preliminary paragraph including the MTO problem statement and SO algorithm.
The proposed algorithm is detailed in paragraph 3, then experimental results are presented and discussed.
2) The MTSO additional complexity needed for the knowledge transfer and and the decision process is not addressed in the paper, authors should estimate the Time/complexity of their algorithm toward related proposals.
3) it is not clear what motivated the selection of "top fifth of individuals for each task as elite individuals", see algorithm table 4, what makes the number (5) pertinence? and how that specific number is set??!!!
4) Comparative analysis need to be confirmed by a Wilcoxon test, to be sure about the pertinence of the comparisons at first?
5) for multi-task test beds, please refer to the following paper :
Li, Y., Gong, W., & Li, S. (2022, July). Evolutionary constrained multi-task optimization: Benchmark problems and preliminary results. In Proceedings of the Genetic and Evolutionary Computation Conference Companion (pp. 443-446).

6) Related to the planar inverse kinematics problem, please state the problem as for a real industrial application one, and clearly list your tasks to better show the importance of the use of MTO for IK problem.
7) The planar robot arm is trivial case, for which classical multi-constrains evolutionary algorithm showed very pertinent results. Please use a real industrial arm model and clearly justify the MTO...!!!

Experimental design

3) it is not clear what motivated the selection of "top fifth of individuals for each task as elite individuals", see algorithm table 4, what makes the number (5) pertinence? and how that specific number is set??!!!
4) Comparative analysis need to be confirmed by a Wilcoxon test, to be sure about the pertinence of the comparisons at first?
5) for multi-task test beds, please refer to the following paper :
Li, Y., Gong, W., & Li, S. (2022, July). Evolutionary constrained multi-task optimization: Benchmark problems and preliminary results. In Proceedings of the Genetic and Evolutionary Computation Conference Companion (pp. 443-446).

Validity of the findings

6) Related to the planar inverse kinematics problem, please state the problem as for a real industrial application one, and clearly list your tasks to better show the importance of the use of MTO for IK problem.
7) The planar robot arm is trivial case, for which classical multi-constrains evolutionary algorithm showed very pertinent results. Please use a real industrial arm model and clearly justify the MTO...!!!

---

## Round 0.2 · accepted · Accept

Reviewer 2 was contacted many times, but has not responded to the invitation.

All the reviewers' comments have been addressed carefully and sufficiently, the revisions are rational from my point of view. Therefore, I think the current version of the paper can be accepted.

·

Basic reporting

No comment. The article meets all basic reporting requirements with clear professional English, sufficient literature references and background context, appropriate structure with figures and tables, self-contained results addressing the hypotheses, and formal results with clear definitions and proofs.

Experimental design

No comment. The experimental design meets all requirements with original primary research within the journal's scope, well-defined research questions addressing a clear knowledge gap, rigorous technical and ethical investigation standards, and sufficiently detailed methods for replication.

Validity of the findings

No comment. The findings demonstrate appropriate validity with clear rationale for replication benefits, robust and statistically sound underlying data, and well-stated conclusions that are properly linked to the research question and supported by the results.

Additional comments

No additional comments needed. The authors have thoroughly addressed all previous concerns and suggestions, making substantial improvements to the manuscript including:

Enhanced English language clarity and readability throughout
Added comprehensive error bar analysis in convergence plots
Expanded experimental validation with noisy functions and constrained optimization problems
Included detailed parameter sensitivity analyses
Added theoretical foundations including complexity analysis and convergence properties
Expanded real-world applications beyond PKACP
Provided clear justification for parameter selections
Enhanced knowledge transfer mechanism documentation and analysis

The revised manuscript now presents a more complete and rigorous treatment of the MTSO algorithm.